# Gastro-Protective Effect of Fermented Soybean (*Glycine max* (L.) Merr.) in a Rat Model of Ethanol/HCl-Induced Gastric Injury

**DOI:** 10.3390/nu14102079

**Published:** 2022-05-16

**Authors:** Minhee Lee, Dakyung Kim, Hyunji Kim, Sukyung Jo, Ok-Kyung Kim, Jeongmin Lee

**Affiliations:** 1Department of Medical Nutrition, Kyung Hee University, Yongin 17104, Korea; miniclsrn@khu.ac.kr (M.L.); k4kyung@naver.com (D.K.); 2Cosmax NBT, Inc., Seongnam-si 13486, Korea; hyunji.kim@cosmaxnbt.com; 3Bision Corporation, Seoul 05854, Korea; skjo@bision.co.kr; 4Division of Food and Nutrition and Human Ecology Research Institute, Chonnam National University, Gwangju 61186, Korea; 5Research Institute of Clinical Nutrition, Kyung Hee University, Seoul 02447, Korea

**Keywords:** *Glycine max* (L.) Merr., gastric damage, gastric ulcer

## Abstract

The present research purposed to examine the gastro-protective effect of *Glycine max* (L.) Merr. fermented using Lactobacillus delbrueckii ssp. delbrueckii Rosell-187 (Gastro-AD^®^) on ethanol/HCl-induced gastric damage, specifically on gastric acid secretion. After oral supplementation of Gastro-AD^®^ to Sprague–Dawley (SD) rats with ethanol/HCl-induced gastric damage, we determined that Gastro-AD^®^ attenuated the gastric mucosal lesion, hemorrhage and gastric acid secretion induced by ethanol/HCl. In addition, we observed that the Gastro-AD^®^ treatment increased the serum prostaglandin E2 level and decreased the levels of gastric acid secretion-related receptors in both gastric tissues and primary gastric parietal cells. Furthermore, it decreased the levels of inflammatory factors, including serum histamine and expression of p-IκB, p-p65, iNOS and COX-2 and the activity of apoptotic signaling pathways, including those involving p-JNK, Bcl2/Bax, Fas, FADD, caspase-8 and caspase-3, in the stomach of the ethanol/HCl-treated rats. Thus, we suggest that Gastro-AD^®^ supplementation may reduce ethanol/HCl-induced gastric acid secretion and prevent gastric injury.

## 1. Introduction

Gastritis is a general term used for inflammatory diseases caused by an imbalance between the gastric mucosal offensive and protective factors in the stomach. It is a common disease that can progress to chronic gastritis, gastric ulcer and gastric carcinoma [1]. Common causes of gastritis are the prolonged use of non-steroidal anti-inflammatory drugs or excessive alcohol consumption, and it can also develop from traumatic injury, bacterial infection (primarily *Helicobacter pylori* infection) and stress. The symptoms of gastritis can include indigestion, burning pain, nausea and vomiting [1,2]. 

Excessive alcohol consumption causes gastric acid secretion and pathological changes, including gastric edema and bleeding, by interrupting the gastric mucosal defense system. Ethanol directly and dose-dependently injures the gastric mucosal barrier by stimulating gastric acid secretion and exacerbating gastric acid-induced gastric mucosal damage [3,4]. The pathways of gastric acid secretion are mediated by histamine-induced histamine 2 receptor (H2r), acetylcholine-induced muscarinic acetylcholine receptor M3 (M3r) and gastrin-induced cholecystokinin 2 receptor (CCK2r). Activation of these receptors increases the intracellular cAMP and Ca^2+^ concentrations, activates H^+^/K^+^ ATPase, which is a proton pump present in the cell membrane, and eventually increases HCl secretion, thereby aggravating the damage to the gastrointestinal mucosa [5,6]. Aihara et al. [7] demonstrated that ethanol-induced gastric acid secretion was significantly inhibited by a treatment with an H2r antagonist, a CCK2r antagonist and an acid pump inhibitor, suggesting that alcoholic beverages induce gastric acid secretion.

Soybean (*Glycine max* (L.) Merr.) is one of the main plant foods that is used as a source of protein in East Asian countries. However, fermented soybean has more beneficial functions than unfermented soybean [8]. Fermentation causes different physical and chemical changes in soybean according to the different organisms and fermentation conditions used [9]. Fatani et al. [10] showed that soybean fermented using *Lactobacillus delbrueckii* ssp. delbrueckii Rosell-187 suppressed heartburn occurrence in individuals who experienced mild or moderate heartburn. In this investigation, we demonstrated the gastro-protective effect of fermented *Glycine max* (L.) Merr. using *Lactobacillus delbrueckii* ssp. delbrueckii Rosell-187 on ethanol/HCl-induced gastric damage, specifically on gastric acid secretion in rats and cells. 

## 2. Materials and Methods

### 2.1. Gastro-AD^®^

*Glycine max* (L.) Merr. was fermented using *Lactobacillus delbrueckii* ssp. delbrueckii Rosell-187 (Gastro-AD^®^). Gastro-AD^®^ was provided by Lallemand Inc. (Grenaa, Denmark). 

### 2.2. Animals and Treatments

Sprague Dawley (SD) rats (male, seven-week-old, 226.50 ± 11.36 g of body weight, b.w.) were housed in cages under automatically controlled humidity (about 50%), lighting (12:12-h light-dark cycle) and temperature (22 ± 2 °C) conditions. The rats were acclimatized for a week before the experiment and then divided into six groups (*n* = 8): saline (NC); gastric inflammation + saline (control, C); gastric inflammation + 30 mg/kg b.w. of rebamipide (positive control, PC); gastric inflammation + 50 mg/kg b.w. of Gastro-AD^®^ (50); gastric inflammation + 100 mg/kg b.w. of Gastro-AD^®^ (100); and gastric inflammation +200 mg/kg b.w. of Gastro-AD^®^ (200). Doses of Gastro-AD^®^ were decided on the basis of a clinical test dose range of Gastro-AD^®^. We calculated using the HED convertation method in the previous papers, in which a human supplement dose was converted to rats [11,12]. The clinical dose of Gastro-AD^®^ ranges from 500 mg to 3000 mg/day [10,13], which for the converted dose of Gastro-AD^®^ ranges from 51.7 mg to 310 mg/kg/day in rats. After oral administration of 1 mL of rebamipide or Gastro-AD^®^ for 14 days, 1 mL of rebamipide or Gastro-AD with 1 mL of 40% ethanol was administrated for 7 days. At the end of 21 days, 2 h after administration of rebamipide or Gastro-AD^®^, acute gastric inflammation was induced via oral administration of 1 mL of 70% ethanol and 0.15 mM HCl to the rats of the C, PC, 50, 100 and 200 groups and then the rats were sacrificed 1 h later. The gastric juice, stomach and serum (via orbital venipuncture) were gathered for analysis. The Institutional Animal Care and Use Committee of Kyung Hee University (KHUASP-21-170) approved the animal study protocol.

### 2.3. Measurement of Gastric Juice Secretion and pH

After the rats were sacrificed, their stomachs were removed and the gastric contents were collected and centrifuged at 3000× *g* for 30 min. The amount of gastric juice was analyzed by measuring the supernatant volume and then the pH of the gastric juice was measured using a pH meter (Thermo Fisher Scientific, Waltham, MA, USA).

### 2.4. Measurement of the Gastric Ulcerative Lesions and Ulceration Index

Each stomach was rinsed with ice-cold 0.9% NaCl and opened along the curvature. Each gastric lesion area was determined using ImageJ processing software (NIH, Bethesda, MD, USA). The lesion area per stomach was calculated using the following equation: Ulcer index (%) = (gastric damage area of each rat/gastric mucosal area of each rat) × 100.

### 2.5. Hematoxylin and Eosin (H&E) Staining

The methods for H&E staining of stomach tissues and the histopathological score index of gastric inflammation were described in a previous study [14].

### 2.6. Serum Histamine and PGE_2_

The levels of serum histamine and PGE_2_ were assessed using quantification kits (R&D system, MSP, MN, USA).

### 2.7. Primary Gastric Parietal Cell Culture

The methods for isolation and culture of the primary gastric parietal cell were described in a previous study [14].

### 2.8. Determination of the cAMP Level in Primary Gastric Parietal Cellsii

The amount of cAMP was determined using a cAMP direct immunoassay kit (Biovision, Milpitas, CA, USA).

### 2.9. Real-Time Polymerase Chain Reaction (RT-PCR)

The methods for total RNA extraction from stomach tissue, cDNA synthesis and real-time PCR were described in a previous study [14]. The primer pairs were described in previous research [14]. Data analysis was performed using the CFX Maestro™ Analysis Software (Bio-Rad).

### 2.10. Western Blot Analysis

The method for protein extraction from stomach tissue and western blot analysis were described in a previous study [14]. We used p-p65, p-IκB, CD95, FADD, p-JNK, BAX, Bcl-2, COX2, cleaved-caspase-8, cleaved-caspase-3 and beta-actin antibodies (Cell signaling, 1:1000) and anti-rabbit IgG HRP-linked antibody (1:5000, Cell Signaling Technology, Inc.). The protein bands were marked using EzWestLumi plus (ATTO & Rise Co., Tokyo, Japan) detection reagents and developed using Ez-Capture II (ATTO & Rise Co., Tokyo, Japan). The bands were quantified using CS Analyzer 3.0 software (ATTO & Rise Co., Tokyo, Japan).

### 2.11. Statistical Analysis

All results are expressed as mean ± standard deviation (S.D.). Statistical analysis was performed using one-way ANOVA and the SPSS statistical procedures for Windows (SPSS PASW Statistic 22.0, SPSS Inc. Chicago, IL, USA). Duncan’s multiple range test was used to analyze the differences among groups. Statistical differences were considered significant for a *p* < 0.05.

## 3. Results

### 3.1. Effect of Gastro-AD^®^ on the Gastric Mucosal Damage and Gastric Acid Secretion Induced by Ethanol/HCl Treatment in SD Rats

We discovered that compared to those induced by the treatment with only saline (NC), the ethanol/HCl treatment (C) induced morphological and histological changes, including hemorrhagic injury lesions and epithelial disruption, in the gastric mucosa (Figure 1A). The ulcer index and histopathological score were significantly higher in the ethanol/HCl-treated rats (C) than in normal rats (NC) (Figure 1B,C). Additionally, the gastric acid volume was significantly higher and the gastric acid pH was significantly lower in the ethanol/HCl-treated rats than in normal rats (Figure 1D,E). However, compared to those of ethanol/HCl-treated rats, rebamipide (PC) and Gastro-AD^®^ 100 and 200 mg/kg b.w. supplementation in ethanol/HCl-treated rats decreased the ulcer index, the histopathological score and the gastric acid volume and increased the gastric acid pH (Figure 2A–D) (*p* < 0.05).

### 3.2. Effect of Gastro-AD^®^ on the Expression of Gastric Acid Secretion-Related Receptors in Gastric Damage Induced by Ethanol/HCl Treatment in SD Rats

The mRNA expression levels of H2r, CCK2r, M3r and H^+^/K^+^
*ATPase* in the stomach of rats treated with ethanol/HCl were elevated compared to those in the rats only treated with saline. When compared with the mRNA levels in the ethanol/HCl treatment control group rats, the rats of the groups with rebamipide and Gastro-AD^®^ supplementation had significantly lower mRNA levels of H2r, CCK2r, M3r and H^+^/K^+^ ATPase in the stomach (Figure 2) (*p* < 0.05).

### 3.3. Effect of Gastro-AD^®^ on the Expression of Gastric Acid Secretion-Related Receptors in Histamine-Treated Primary Gastric Parietal Cells

Compared to that in normal cells, histamine treatment of primary gastric parietal cells increased the mRNA expression levels of *H2r, CCK2r, M3r* and *H^+^/K^+^ ATPase*. Moreover, compared with those of the control group cells, the treatment with Gastro-AD^®^ in histamine-treated primary gastric parietal cells suppressed the mRNA expression of H2r, CCK2r, M3r and H^+^/K^+^ ATPase and decreased the intracellular cAMP level (Figure 3) (*p* < 0.05).

### 3.4. Effect of Gastro-AD^®^ on the Inflammatory Factors in Gastric Damage Induced by Ethanol/HCl Treatment in SD Rats

The protein expression levels of p-IκB, p-p65, iNOS and COX-2 in the stomach of ethanol/HCl-treated rats were significantly higher than in the stomach of the control group rats. However, compared with those of the control group, the rebamipide and Gastro-AD^®^ supplementation groups showed a significant decrease in the protein expression levels of p-IκB, p-p65, iNOS and COX-2 (Figure 4A,B) (*p* < 0.05).

Figure 4C shows that the serum histamine level was higher in the ethanol/HCl treatment control group rats than in the normal control group rats. Ethanol/HCl-treated rats supplemented with rebamipide and Gastro-AD^®^ showed a significant decrease in the serum histamine. We determined that the ethanol/HCl treatment decreased the serum prostaglandin E2 (PGE_2_) level, but rebamipide and Gastro-AD^®^ supplementation caused a significant increase in serum PGE_2_ level compared with those of the control group (Figure 4D) (*p* < 0.05).

### 3.5. Effect of Gastro-AD^®^ on the Apoptosis Factors in Gastric Damage Induced by Ethanol/HCl Treatment in SD Rats

We measured the expression levels of the proteins in the apoptotic signaling pathway (p-JNK, Bcl2/Bax, Fas, FADD, caspase-8 and caspase-3) in the gastric tissue of ethanol/HCl-treated rats. We determined that compared with those of the control group, the ethanol/HCl treatment increased the protein expression levels of the apoptotic signaling pathway factors, but rebamipide and Gastro-AD^®^ supplementation caused a significant decrease in protein expression levels (Figure 5) (*p* < 0.05). Appendix A is expanded and uncropped western blot panels from Figure 4 and Figure 5.

## 4. Discussion

The main function of the gastric system is to store, grind and dispense food for further digestion and absorption under neural and hormonal control [15]. In addition, gastric acid prevents bacterial overgrowth and enteric infections, supports protein digestion and facilitates the absorption of vitamin B_12_, iron and calcium [16]. However, excess gastric acid secretion can cause abdominal discomfort, nausea, vomiting, heartburn and diarrhea; such excess secretion can also increase the risk of developing gastrointestinal bleeding, peptic ulcers and gastroesophageal reflux disease [17]. In this study, we determined the gastro-protective effect of Gastro-AD^®^ on gastric injury caused by ethanol/HCl-induced gastric acid secretion in SD rats. We determined that ethanol/HCl-induced gastric injury caused pathological changes, including mucosal edema, gastric bleeding and epithelial desquamation in the stomach, along with gastric acid secretion. However, we observed that the Gastro-AD^®^ treatment suppressed the morphological and histological damage and gastric acid secretion induced by ethanol/HCl treatment.

We showed that the expression levels of gastric acid secretion-mediated receptors were elevated after the ethanol/HCl treatment. Gastric acid is secreted by three stimuli: histamine (released from ECL cells), acetylcholine (released from the postganglionic enteric neurons) and gastrin (released from G cells) in the parietal cells. The binding of acetylcholine to the M3r and the binding of gastrin to CCK2r involve an increase in the intracellular *Ca*^2+^ level in parietal cell. The binding of histamine to H2r stimulates the activation of adenylate cyclase and the generation of cAMP. An increased intracellular Ca^2+^ level and the generation of cAMP stimulate H^+^/K^+^ ATPase, also called a proton pump, which secretes H^+^ into the gastric lumen [5,18]. In the present study, we showed that the ethanol/HCl treatment increased the level of histamine and the mRNA expression levels of H2r, CCK2r, M3r and H^+^/K^+^ ATPase in rats. Additionally, histamine-treated primary gastric parietal cells showed increased mRNA expression levels of H2r, CCK2r, M3r and H^+^/K^+^ ATPase and a higher intracellular cAMP level. However, the Gastro-AD^®^ treatment in both rats and cells suppressed the mRNA expression of H2r, CCK2r, M3r and H^+^/K^+^ ATPase. These results suggest that Gastro-AD^®^ decreased the ethanol/HCl-induced gastric acid secretion by directly suppressing the expression of the gastric acid secretion-mediated receptors in gastric parietal cells.

Next, we demonstrated the effects of Gastro-AD^®^ against alcohol supplementation-induced gastric inflammation. We confirmed increased expression levels of phosphorylated IκB, p65, iNOS and COX-2 in ethanol/HCl-induced gastric damage conditions. However, compared with the ethanol/HCl treatment, the Gastro-AD^®^ treatment reduced the expression levels of phosphorylated IκB, p65, iNOS and COX-2. Jung et al. [19] reported that the oral administration of an extract of fermented *Glycine max* reduced the levels of inflammatory factors, including iNOS, in the skin of atopic dermatitis-like NC/Nga mice. Kim et al. [20] discovered that the antioxidant ability was enhanced after fermentation using *Glycine max*, and that of fermented *Glycine max* had anti-atherosclerosis effects by suppressing the inflammatory cytokines regulated by NF-kB signaling. These findings, along with our results, suggest that Gastro-AD^®^ might be strongly associated with the relief of ethanol/HCl-induced inflammation development occurring during fermentation.

Several studies have established the anti-proliferative effects of fermented *Glycine max* in cancer cell lines through apoptosis induction [21,22,23]. In our study, we showed that the ethanol/HCl treatment increased the expression levels of proteins of the apoptotic signaling pathway, including p-JNK, Bcl2/Bax, Fas, FADD, caspase-8 and caspase-3. However, the Gastro-AD^®^ treatment suppressed the apoptotic signaling pathway in the gastric tissues from ethanol/HCl-induced rats. Moreover, Gastro-AD^®^ treatment caused a significant increase in serum PGE_2_ level. PGE_2_ is known to have a strong protective effect on the gastric mucosa [24]. Hoshino et al. [25] showed that PGE_2_ inhibited the release of cytochrome c from mitochondria in ethanol- treated gastric mucosal cells and of EP receptors involved in the PGE_2_-mediated protection of cells from ethanol-induced apoptosis. Thus, we suggest that fermented *Glycine max* has a protective effect by regulating the PGE_2_ level and apoptosis according to the state of the cell.

## 5. Conclusions

In conclusion, our results reveal that the oral administration of Gastro-AD^®^ has protective effects on the gastric injury induced by ethanol/HCl treatment in SD rats. This effect was associated with the suppression of the expression of gastric acid secretion-mediated receptors, inflammation and apoptosis in ethanol/HCl-treated rats. These data suggest that Gastro-AD^®^ supplementation may prevent gastric injury through the suppression of gastric acid secretion.

## Figures and Tables

**Figure 1 nutrients-14-02079-f001:**
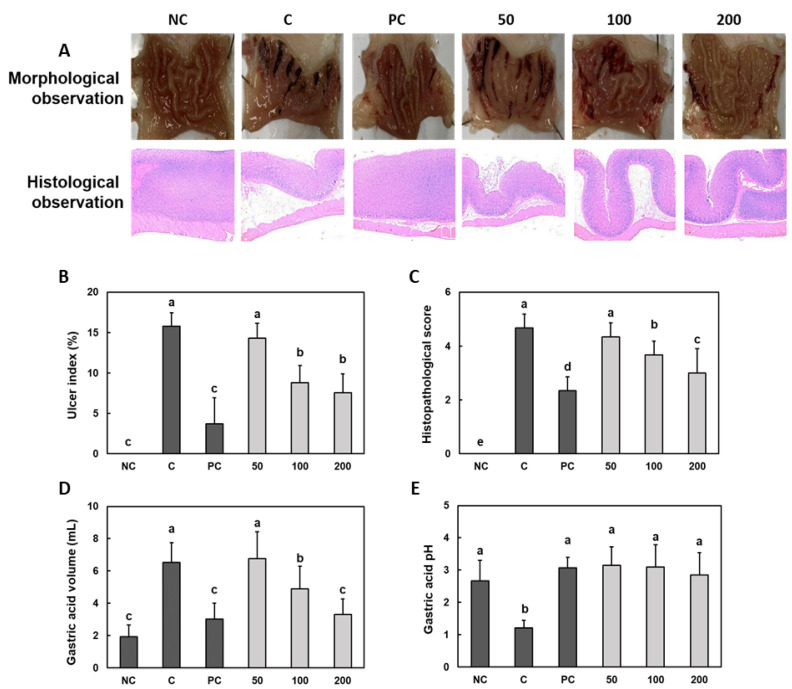
Effect of Gastro-AD^®^ on the gastric mucosal injury and gastric acid secretion induced by ethanol/HCl treatment in SD rats. (**A**) Representative images of the morphological and histological observations obtained after H&E staining in each group. The (**B**) gastric ulcer index, (**C**) histopathological score, (**D**) gastric acid volume and (**E**) gastric acid pH values. Normal control (NC; −ethanol/HCl); ethanol/HCl control (C); ethanol/HCl with 30 mg/kg b.w. of rebamipide (positive control; PC); ethanol/HCl with 50 mg/kg b.w. of Gastro-AD^®^ (50); ethanol/HCl with 100 mg/kg b.w. of Gastro-AD^®^ (100); and ethanol/HCl with 200 mg/kg b.w. of Gastro-AD^®^ (200). Values are presented as mean ± SD (*n* = 8). Different letters indicate a significant difference, with *p* < 0.05, as determined using Duncan’s multiple range test.

**Figure 2 nutrients-14-02079-f002:**
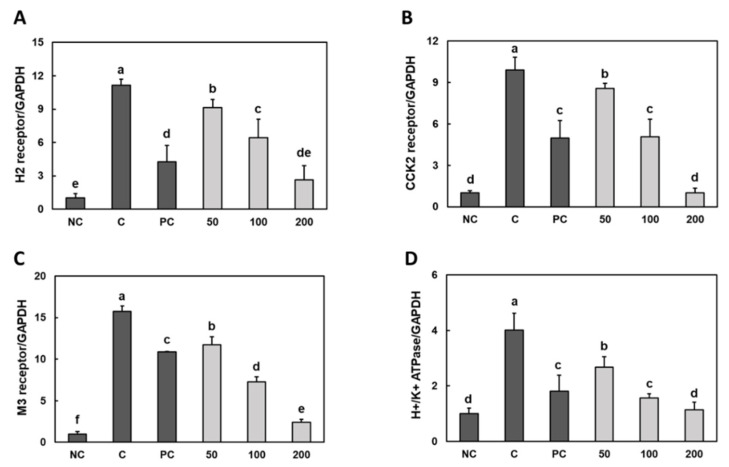
Effect of Gastro-AD^®^ on the expression of gastric acid secretion-related receptors in gastric damage induced by ethanol/HCl treatment in SD rats. The mRNAs levels of (**A**) histamine-induced histamine 2 receptor (H2r), (**B**) gastrin-induced cholecystokinin 2 receptor (CCK2r), (**C**) acetylcholine-induced muscarinic acetylcholine receptor M3 (M3r) and (**D**) H^+^/K^+^ ATPase. Normal control (NC; −ethanol/HCl); ethanol/HCl control (C); ethanol/HCl with 30 mg/kg b.w. of rebamipide (positive control; PC); ethanol/HCl with 50 mg/kg b.w. of Gastro-AD^®^ (50); ethanol/HCl with 100 mg/kg b.w. of Gastro-AD^®^ (100); and ethanol/HCl with 200 mg/kg b.w. of Gastro-AD^®^ (200). Values are presented as mean ± SD (*n* = 8). Different letters indicate a significant difference, with *p* < 0.05, as determined using Duncan’s multiple range test.

**Figure 3 nutrients-14-02079-f003:**
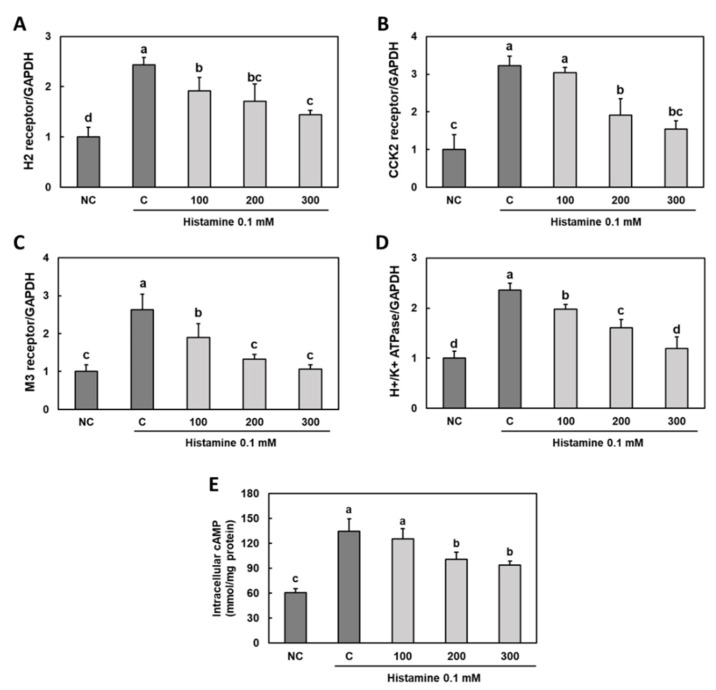
Effect of Gastro-AD^®^ on the expression of gastric acid secretion-related receptors in histamine-treated primary gastric parietal cells. The mRNA levels of (**A**) histamine-induced histamine 2 receptor (H2r), (**B**) gastrin-induced cholecystokinin 2 receptor (CCK2r), (**C**) acetylcholine-induced muscarinic acetylcholine receptor M3 (M3r), (**D**) H^+^/K^+^ ATPase and (**E**) intracellular cAMP. Normal control (NC; −0.1 mM histamine); 0.1 mM histamine control (C); 0.1 mM histamine with 100 μg/mL of Gastro-AD^®^ (100); 0.1 mM histamine with 200 μg/mL of Gastro-AD^®^ (200); and 0.1 mM histamine with 300 μg/mL of Gastro-AD^®^ (300). Values are presented as mean ± SD (*n* = 3). Different letters indicate a significant difference, with *p* < 0.05, as determined using Duncan’s multiple range test.

**Figure 4 nutrients-14-02079-f004:**
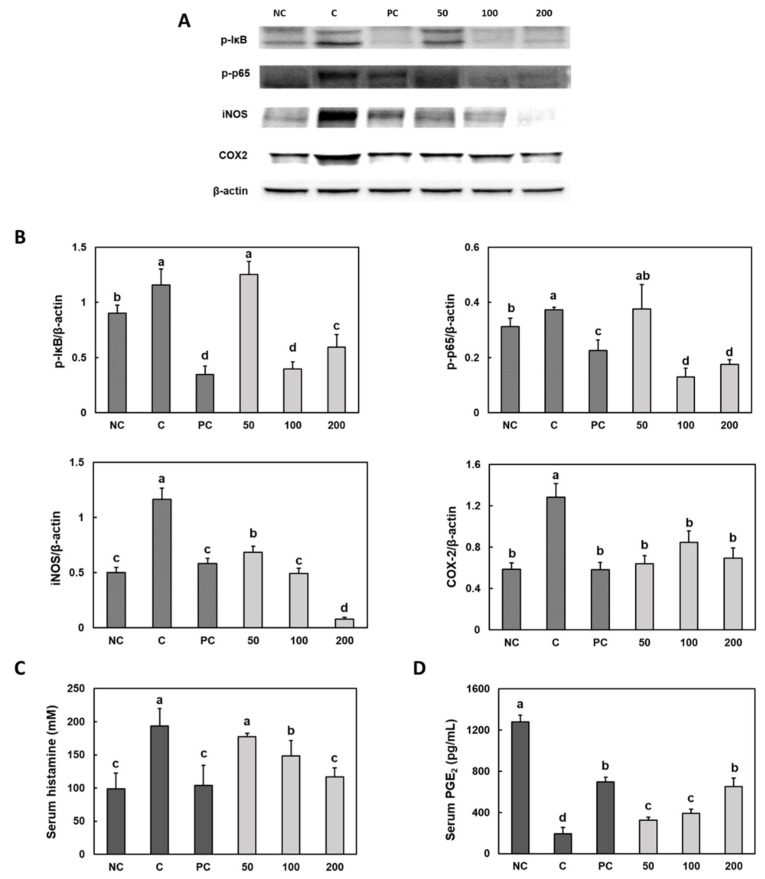
Effect of Gastro-AD^®^ on the inflammatory factors and PGE_2_ in gastric damage induced by ethanol/HCl treatment in SD rats. The protein expression of inflammatory factors (**A**), band image; (**B**), quantification of bands) and the levels of serum (**C**) histamine and (**D**) PGE_2_. Normal control (NC; −ethanol/HCl); ethanol/HCl control (C); ethanol/HCl with 30 mg/kg b.w. of rebamipide (positive control; PC); ethanol/HCl with 50 mg/kg b.w. of Gastro-AD^®^ (50); ethanol/HCl with 100 mg/kg b.w. of Gastro-AD^®^ (100); and ethanol/HCl with 200 mg/kg b.w. of Gastro-AD^®^ (200). Values are presented as mean ± SD (*n* = 8). Different letters indicate a significant difference, with *p* < 0.05, as determined using Duncan’s multiple range test.

**Figure 5 nutrients-14-02079-f005:**
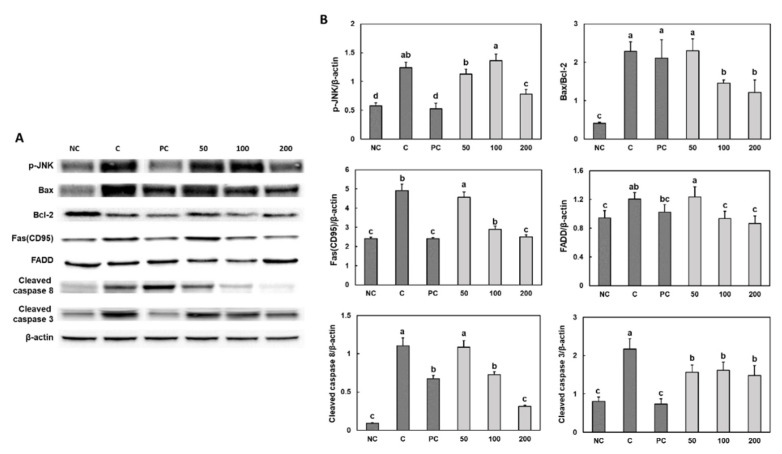
Effect of Gastro-AD^®^ on the apoptosis-associated factors (**A**), band image; (**B**), quantification of bands) in gastric damage induced by ethanol/HCl treatment in SD rats. Normal control (NC; −ethanol/HCl); ethanol/HCl control (C); ethanol/HCl with 30 mg/kg b.w. of rebamipide (positive control; PC); ethanol/HCl with 50 mg/kg b.w. of Gastro-AD^®^ (50); ethanol/HCl with 100 mg/kg b.w. of Gastro-AD^®^ (100); and ethanol/HCl with 200 mg/kg b.w. of Gastro-AD^®^ (200). Values are presented as mean ± SD (*n* = 8). Different letters indicate a significant difference, with *p* < 0.05, as determined using Duncan’s multiple range test.

## Data Availability

Not applicable.

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
