# Peer review of "Gastro-Protective Effect of Fermented Soybean (Glycine max (L.) Merr.) in a Rat Model of Ethanol/HCl-Induced Gastric Injury"

_nutrients, 2022, doi:10.3390/nu14102079_

Round 1

Reviewer 1 Report

In the present study, Lee et al investigated the gastro-protective effect of Glycine max (L.) Merr. Fermented using Lactobacillus ssp. Delbrueckii-187 (Gastro-AD) on ethanol/Hcl-induced gastric damage in rats. My comments are as follows:

  • What is the basis of selecting Gasto-AD dose range (50-200 mg/kg b.w.). Is there any correlation with its prescribed human dose?? Please include its justification in the manuscript.
  • H&E staining images in Fig. 1A are of very low resolution and it’s hard to see any cellular composition changes. Please include high resolution images.
  • Regarding western blots figures, blots of different protein should be aligned in a proper way. Are different proteins were looked at in the same blot or in different blots?? If more than one blot has been used in the same figure, separate β- actin corresponding to each blot should be shown. Also, please refrain from too much processing of blots, e.g. Each lane in iNOS blot is having different background. Besides this, please provide original uncut full length blot for each protein.
  • Effect of Gastro-AD on various enzymes (SOD, Catalase, GPX etc.) involved in oxidative stress in the stomach tissue is worth exploring to have better understanding of the mechanism of action.

Author Response

Thanks for your careful review.

  • We wrote Gastro-AD dose selecting reason and changed H&E staining images resolution.
  • We lost some original uncut full length photo (blot), so we provide all the photo (blot) we had.
  • In a future study, We will be to investigate with the measurement of biomarkers related to oxidative stress.

Reviewer 2 Report

           The structure of manuscript has the commonly required criteria. The topic of presented work is very actual. Gastritis is a general term used for inflammatory diseases caused by an imbalance between the gastric mucosal offensive and defensive factors in the stomach. It is a common disease that can progress to chronic gastritis, gastric ulcer, and gastric carcinoma.

            In the present study, authors investigated the gastro-protective effect of Glycine max (L.) Merr. fermented using Lactobacillus delbrueckii ssp. delbrueckii Rosell-187 (Gastro-AD®) on etha-15 nol/HCl-induced gastric damage, specifically on gastric acid secretion. The research work follows several aims and interesting parametric studies. After oral administration of Gastro-AD® to Sprague–Dawley (SD) rats with ethanol/HCl-induced gastric damage, authors determined that Gastro-AD® attenuated the gastric mucosal injury, hemorrhage, and gastric acid secretion induced by ethanol/HCl. The authors observed that the Gastro-AD® treatment increased the serum prostaglandin E2 level and decreased the levels of gastric acid secretion-related receptors in both gastric tissues and primary gastric parietal cells. The results are documented in graphs that present the review of the obtained data. 

            The citations are well-chosen and relevant and their format respects usual standards. The conclusion summarizes the author´s results.

Author Response

Thanks for your careful review.

Round 2

Reviewer 1 Report

My comments for the revised manuscript are as follow:

  • Although authors stated now that human equivalent dose was used but still missing the criteria and method used for conversion of human dose to the animal dose used in the present study.
  • There are few grammatical mistakes and spelling mistakes in the newly added text (e.g. Dose has been mentioned as does).
  • H&E staining images are still not in high resolution as requested earlier
  • Authors have not addressed the previously raised serious concerns about the western blots used in the current manuscript and stated loss of original blots as requested earlier. Integrity of the data warrants further confirmation.

Author Response

Thank you for your comments and suggestion again. 
We changed the manuscript as follow;

1. We wrote manuscript that the reason for animal supply concentration determination. 
2. We changed H&E staining images in high resolution again. 
3. We applied the original blot images as the reviewer requested, but we submitted the cropped blot bands. We already cropped protein bands for clean band images when anayzed, thus we don't have full band images. 
